# Empowering AlphaFold2 for protein conformation selective drug discovery with AlphaFold2-RAVE

Xinyu Gu[1,2], Akashnathan Aranganathan[1,3], Pratyush Tiwary[1,2,4]*

[1]Institute for Physical Science and Technology, University of Maryland, College Park, United States; [2]University of Maryland Institute for Health Computing, Bethesda, United States; [3]Biophysics Program, University of Maryland, College Park, United States; [4]Department of Chemistry and Biochemistry, University of Maryland, College Park, United States

*For correspondence:
pratyush.tiwary@gmail.com

**Abstract** Small-molecule drug design hinges on obtaining co-crystallized ligand-protein structures. Despite AlphaFold2's strides in protein native structure prediction, its focus on apo structures overlooks ligands and associated holo structures. Moreover, designing selective drugs often benefits from the targeting of diverse metastable conformations. Therefore, direct application of AlphaFold2 models in virtual screening and drug discovery remains tentative. Here, we demonstrate an AlphaFold2-based framework combined with all-atom enhanced sampling molecular dynamics and Induced Fit docking, named AF2RAVE-Glide, to conduct computational model-based small-molecule binding of metastable protein kinase conformations, initiated from protein sequences. We demonstrate the AF2RAVE-Glide workflow on three different mammalian protein kinases and their type I and II inhibitors, with special emphasis on binding of known type II kinase inhibitors which target the metastable classical DFG-out state. These states are not easy to sample from AlphaFold2. Here, we demonstrate how with AF2RAVE these metastable conformations can be sampled for different kinases with high enough accuracy to enable subsequent docking of known type II kinase inhibitors with more than 50% success rates across docking calculations. We believe the protocol should be deployable for other kinases and more proteins generally.

## eLife assessment

This **important** study demonstrates that combining AlphaFold2 with the author's sampling method AF2-RAVE improves protein-ligand docking for three protein kinases and their inhibitors. The evidence is **compelling** and the results will be of interest to researchers who work on computer-aided drug design.

## Introduction

Despite the groundbreaking impact of AlphaFold2 (AF2) (*Jumper et al., 2021*; *Mirdita et al., 2022*) on the computational prediction of the ligand-free protein native apo structures, it appears that the determination of high-quality crystal or cryo-EM ligands-bound holo structures remains irreplaceable in the field of structure-based drug design. When ligands bind, residues within the protein pockets may adjust their side-chain rotamer configurations to optimize contacts with ligands, a phenomenon known as the induced fit effect. Furthermore, thermodynamic fluctuations induce protein dynamics and structural flexibility, leading to rearrangements of side-chains and even large-scale backbone movements that can reveal cryptic pockets (*Amaro, 2019*). These metastable conformations and

associated cryptic pockets can be stabilized upon binding to specific ligands, a phenomenon known as conformational selection. Moreover, due to the similarity of native structures among protein homologs, it has been widely believed that ligands targeting highly diverse metastable conformations should result in better selectivity (*Davis et al., 2011*). It is thus highly desirable to account for metastable protein conformations or states, instead of investigating native states only. Several AF2-based techniques, including reduced multiple sequence alignment (rMSA) AF2 (or MSA subsampling AF2) (*Del Alamo et al., 2022*; *Monteiro da Silva et al., 2024*; *Porter et al., 2023*), AF2-cluster (*Wayment-Steele et al., 2024*), and AlphaFlow (*Jing et al., 2024*), have been devised to generate distinct decoy structures from native states. However, the suitability of these decoys for subsequent docking and virtual screening remains uncertain. Moreover, accurately assigning Boltzmann weights to decoys produced by those methods lacking a direct physical interpretation is challenging. Such a Boltzmann ranking is critical simply because of the explosion in number of decoys that can be hallucinated from AF2 or future generative AI methods, now including AlphaFold3 (*Abramson et al., 2024*).

The AF2RAVE protocol integrates rMSA AF2 and the machine learning-based reweighted autoencoded variational Bayes for enhanced sampling (RAVE) method (*Wang et al., 2019*; *Wang and Tiwary, 2021*; *Mehdi et al., 2024*) to systematically explore metastable states and accurately rank structures using Boltzmann weights (*Vani et al., 2023*). Subsequently, traditional grid-based docking methods or recent generative diffusion models, like DiffDock (*Corso et al., 2022*) and DynamicBind (*Lu et al., 2024*), can be employed to dock ligands with a few top-ranked structures in metastable states, enabling further virtual screening on large ligand libraries. In this work, we chose the docking method Glide XP (*Friesner et al., 2004*; *Halgren et al., 2004*; *Friesner et al., 2006*) and Induced Fit

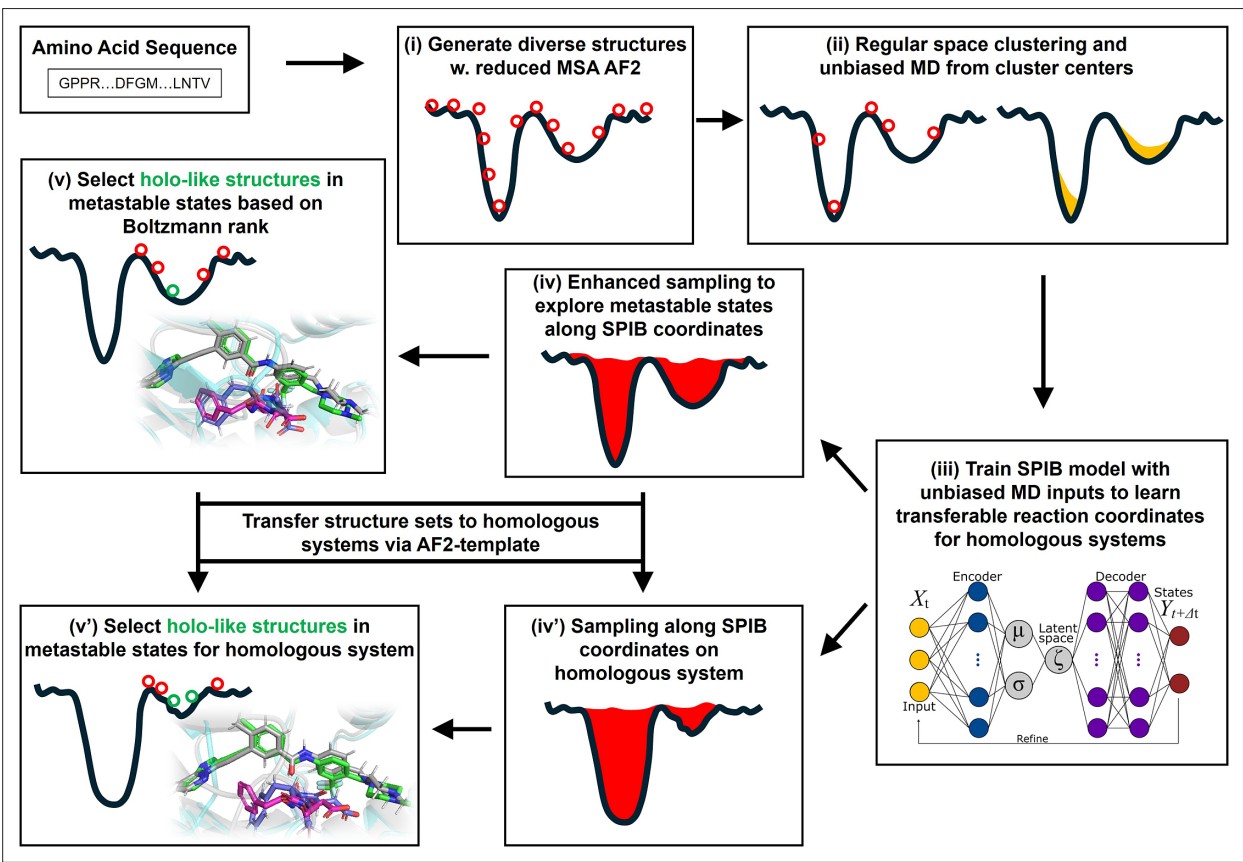

**Figure 1.** A schematic of the AF2RAVE-Glide workflow. A schematic of the AF2RAVE-Glide workflow: (**i**) decoy structures generated by reduced multiple sequence alignment (MSA) AlphaFold2 (AF2), (**ii**) regular space clustering and unbiased molecular dynamics (MD) simulations starting from cluster centers, (**iii**) state predictive information bottleneck model (SPIB, a reweighted autoencoded variational Bayes for enhanced sampling [RAVE] variant) to learn reaction coordinates from unbiased MD, (**iv**) enhanced sampling runs to calculate free energy landscape, (**v**) distinguish holo-like structures from decoys in metastable states based on Boltzmann rank and conduct Glide XP or Induced Fit docking (IFD) on holo-like structures for ligands targeting metastable states. (**iv′**) and (**v′**) The decoy structure set and the learnt SPIB coordinates are transferable to homologous systems.

docking (IFD) (*Sherman et al., 2006a*; *Sherman et al., 2006b*) from the Schrödinger suite (*Maestro, 2023*) as our primary docking method. By combining these two steps, we propose the AF2RAVE-Glide workflow (*Figure 1*) as an innovative approach for small-molecule drug design, initiated from protein sequences. In this workflow, the conformational selection effect is addressed by probing various meta-stable states using AF2RAVE. Glide-IFD is then applied to account for the induced fit effect, refine the holo-like pockets further, and predict the ligand-bound holo structures.

In this work, we demonstrate this AF2RAVE-Glide workflow retrospectively on three different protein kinases and their type I and type II inhibitors. Protein kinases are involved in the regulation of various cellular pathways by catalyzing the hydrolyzing of ATP and transferring the phosphate group to substrate peptides/proteins. Dysfunctions of various kinases are known to cause human pathologies and cancers. The human genome contains about 500 protein kinases which share highly conserved structures in their catalytic ATP-binding pocket, due to the selection pressure toward functional catalysis. This poses significant challenges in developing selective small-molecule ATP-competitive kinase inhibitors, as they must effectively target the intended kinase while avoiding off-target interactions and associated side effects. Research efforts aimed at achieving selectivity have led to the development

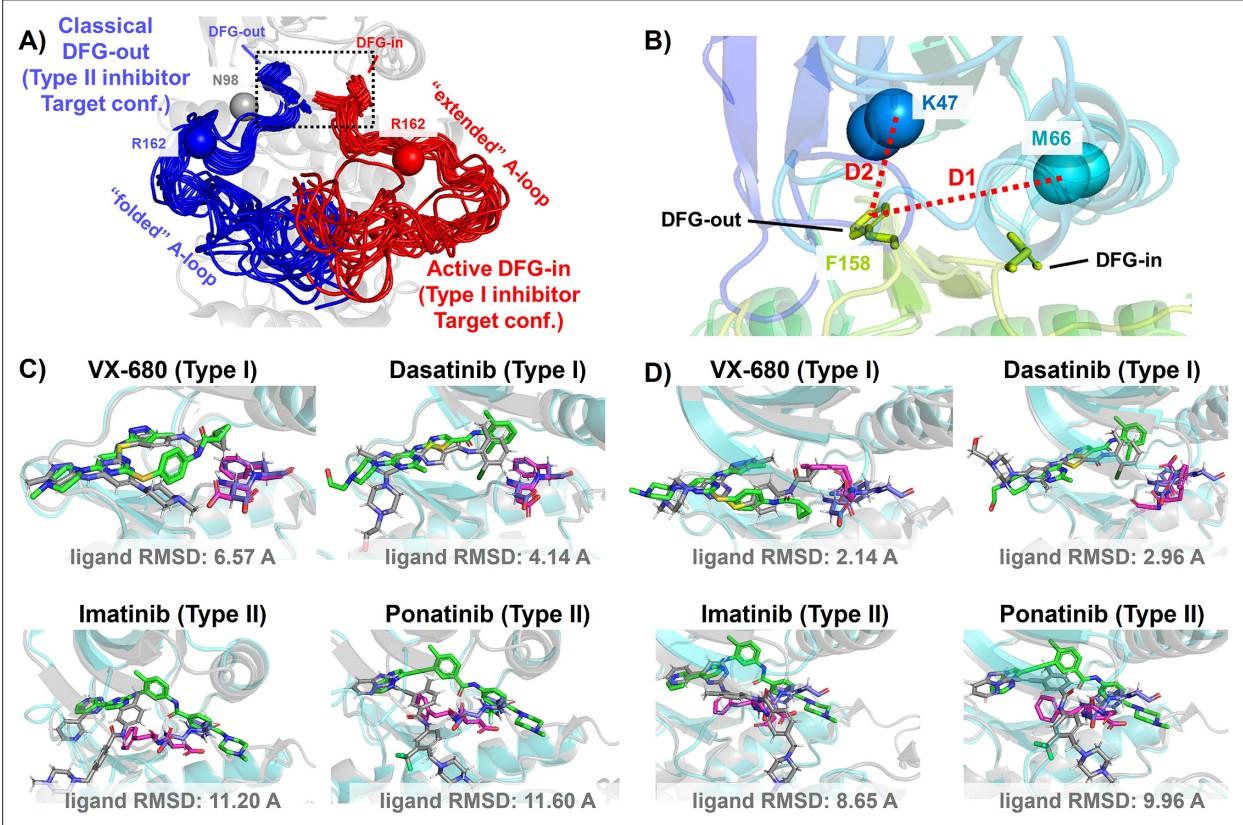

**Figure 2.** DFG-in and DFG-out conformation adopted by DDR1 kinase and their relation with Type I and Type II inhibitors. (**A**) The NMR (nuclear magnetic resonance) structures of Abl1 kinase overlay, comparing the activation loop (A-loop) in the active DFG-in state (red, Protein Data Bank [PDB]: 6XR6) with the classical DFG-out state (blue, PDB: 6XRG). Type I inhibitors target the active DFG-in state, where the DFG motif adopting the DFG-in configuration and the A-loop adopting the 'extended' configuration, while type II inhibitors target the classical DFG-out state, where the DFG motif adopting the DFG-out configuration and the A-loop is 'folded'. The distance between CB atoms of residue N98 (gray bead) and residue R162 (red/blue bead) in Abl1 kinase serves as an order parameter here to illustrate the location of the A-loop. The dashed black block emphasizes the different configurations of the DFG motif in these two states. (**B**) The Dunbrack definition for DFG motif configuration is employed here. The Dunbrack space is delineated by two-order parameters: D1=dist(F158-CZ, M66-CA), D2=dist(F158-CZ, K47-CA). (**C** or **D**) The docking poses with the lowest ligand RMSD for four known kinase inhibitors targeting the Abl1 or DDR1 kinase AlphaFold2 (AF2) structure, generated by Glide XP. Co-crystallized structures are shown as light-cyan cartoons (proteins), green sticks (ligand), and magenta sticks (DFG motif). Docking poses are shown as light-gray cartoons (proteins), gray sticks (ligand), and blue sticks (DFG motif). Comparing with type I inhibitors, AF2 structures of protein kinases fail to dock with type II inhibitors.

The online version of this article includes the following figure supplement(s) for figure 2:

**Figure supplement 1.** The distributions of ligand RMSDs for Glide XP docking poses of DDR1 and type I/type II inhibitors (upper/lower panel).

of four distinct types of kinase inhibitors (*Müller et al., 2015*), including two types of ATP-competitive kinase inhibitors: type I inhibitors bind to the ATP-binding site adopting the active catalytic state, while type II inhibitors target the binding pockets adjacent to the ATP-binding site adopting the inactive state. These two states primarily differ in the configurations of their activation loop (A-loop), which is a flexible loop of approximately 20 residues. In the active state, the A-loop is 'extended' to create a cleft for substrate peptides to bind, while in the inactive state, it is collapsed or 'folded' onto the protein surface, blocking substrate binding. Additionally, in the active state, the three residues 'Asp-Phe-Gly' (DFG motif) at the N-terminus of the A-loop bind to an ATP-binding $Mg^{2+}$ ion, with the Asp side-chain pointing inward to the ATP-binding pocket, while in the inactive state, the Asp side-chain is flipped outward, and the DFG motif adopts the DFG-out conformation (*Modi and Dunbrack, 2019*). Henceforth, we will refer to the type I inhibitor binding state as the active DFG-in state and designate the type II inhibitor binding state as the classical DFG-out state, as previously proposed in the literature (*Gizzio et al., 2022*; *Thakur et al., 2024*; *Gizzio and Thakur, 2024*) (demonstrated in *Figure 2A and B* using Abl1 kinase as an example).

In this work, we investigated three kinases: (i) Abl1, which is targeted by the first clinically approved small-molecule kinase inhibitor imatinib for cancer therapy; (ii) DDR1, a structurally more flexible kinase which is identified as a promiscuous kinase targeted by chemically diverse inhibitors (*Hanson et al., 2019*); and (iii) Src kinase, another crucially important member of the tyrosine kinase family. In the following section, we applied AF2RAVE to enrich holo-like structures adopting the classical DFG-out state from the AF2-generated ensembles of DDR1 and Abl1 kinase, and further validated those holo-like structures by docking them with known type II kinase inhibitors. We initiated our investigation by conducting docking experiments involving both type I and type II inhibitors with the AF2 structures of these two kinases. We observed the incapacity of AF2 structures to effectively dock with type II inhibitors targeting the metastable classical DFG-out state. Subsequently, we employed rMSA AF2, an associative memory-like process that generates diverse structure ensembles for both kinases (*Porter et al., 2023*; *Roney and Ovchinnikov, 2022*), exploring their relatively improved but still limited potential to contain holo-like structures suitable for type II inhibitor binding. Ultimately, our study showcases AF2RAVE's effectiveness in enhancing the generation and selection of holo-like structures in metastable states by integrating AF2-based ensemble generation with physics-based methods.

## Results and discussion
### AF2 structures fail to dock with type II kinase inhibitors

The utility of AF2-generated structures for structure-based drug discovery and virtual screening campaigns has been a subject of controversy and skeptical enthusiasm (*Lyu et al., 2024*; *Holcomb et al., 2023*; *Scardino et al., 2023*; *Díaz-Rovira et al., 2023*). For proteins lacking crystal and cryo-EM structures but with homologous structures available in PDB, such as CDK20 kinase, AF2 demonstrates effectiveness as a homology modeling method for generating initial structures suitable for subsequent virtual screening (*Ren et al., 2023*). Other AF2-based homology modeling method can bias AF2-generated structures toward user-selected template structures with specific druggable conformations (*Sala et al., 2023*). However, a significant drop in the hits enrichment factor during virtual screening has been reported when employing AF2 structures as rigid receptors for docking, compared to using holo PDB structures. This occurs even in cases where the binding pockets of AF2 structures differ only slightly at two to three residues, from those of holo PDBs (*Scardino et al., 2023*; *Díaz-Rovira et al., 2023*). Given that ligands can induce slight relocation and side-chain rotation of pocket residues upon binding, it is important to note that AF2 does not account for this induced fit effect, as it does not encode co-factors like ligands. Therefore, it appears necessary to perform ligand induced fit modeling or relaxation on AF2 structures before engaging in any further structure-based drug design. Molecular dynamics (MD) simulations biased toward adjacent holo-template structure have proved effective in refining apo structures and improving their early enrichment performance (*Guterres et al., 2021*). It has also been demonstrated that AF2 structures can achieve comparable accuracy to crystal holo structures in free energy perturbation (FEP) calculations, by superposing AF2 structures with crystal structures, grafting the co-crystallized ligands onto the AF2 structure, and optimizing the AF2 structure/ligand complex to account for subtle induced fit effects (*Beuming et al., 2022*). AF2 structures, decorated with ligands from template ligand grafting method or known-hits docking method, and

further refined by Schrödinger IFD-MD protocol, exhibit promising performance in early enrichment (*Zhang et al., 2023*) and the prediction of novel ligand/protein complex structures (*Coskun et al., 2024*). Therefore, if large backbone motion is not required, it appears feasible to refine the apo pockets in AF2 structures into holo-like pockets for structure-based drug design. However, when AF2 structures exhibit significant steric clashes with holo ligands, especially in cases ligands targeting metastable states, direct application of out-of-the-box AF2 structures in docking methods for virtual screening and early enrichment may pose more challenges. The AF2-predicted kinase structures predominantly exhibit the DFG-in state, with over 95% of human kinases predicted in this conformation (*Modi and Dunbrack, 2022*; *Al-Masri et al., 2023*). Significant A-loop motion and backbone flipping of the DFG motif are necessary to transition from the AF2-predicted DFG-in state to holo-like states, for type II ligands targeting the classical DFG-out state. As a result, AF2 structures of Abl1/DDR1 kinases exhibit superior performance when docking with type I inhibitors (achieving a minimum ligand RMSD of 2.14 Å) compared to docking with type II inhibitors using Glide XP (*Figure 2C and D*). Even with Glide's IFD (*Figure 3—figure supplement 4*) and the highly side-chain clashes forgiving docking method DiffDock (*Figure 3—figure supplement 6*), AF2 structures struggle to dock with these metastable state-targeting ligands (type II inhibitors), with ligand RMSDs above 8 Å across all docking poses from all three docking methods.

## Holo-like metastable structures may be present among decoys generated from rMSA AF2

AF2-based methods can achieve structural diversity by introducing dropouts in MSA inputs stochastically (rMSA AF2) or in a clustering manner (AF2-cluster). Additionally, models modified from the AF2 framework, such as the flow-match generative model AlphaFlow (*Jing et al., 2024*), have been developed to explore the diversity of conformational space. Similar protocols to rMSA AF2 have demonstrated the potential to address the induced fit effect by generating diverse structures at the binding pocket, ranging from closed apo pockets to opened holo-like pockets (*Meller et al., 2023*). Other investigations have also indicated that larger backbone motions, such as DFG motif backbone flipping and A-loop movement in at least some protein kinases, can be captured by the rMSA AF2 ensemble, although the distributions of conformations deviate significantly from the correct Boltzmann distributions (*Vani et al., 2024*; *Monteiro da Silva et al., 2024*).

In this section, we utilized rMSA AF2 to generate 1280 diverse structures for Abl1 kinase or DDR1 kinase: 640 for MSAs of depth 16 and 32. See Supplementary Material for information regarding Src kinase. Following a filtering step based on RMSD from corresponding AF2 structures, 1198 and 1147 structures remain for Abl1 and DDR1, respectively (*Figure 3—figure supplement 2*). As shown in *Figure 3A and B*, for Abl1 kinase, only 4 out of 1198 structures have a folded A-loop, when using a distance cutoff of 15 Å between CB atoms of N98 and R162 in Abl1. However, for DDR1 kinase, 124 out of 1147 rMSA AF2 structures exhibit a folded A-loop, when using a salt bridge distance cutoff of 10 Å between the aligned residue pairs in DDR1 (E110 and R191). We then clustered the A-loop folded structures in the Dunbrack space. For Abl1, only two clusters were identified: one adopts the DFG-in state, and the other adopts the DFG-inter state, referring to an intermediate conformation during the backbone flipping from DFG-in to DFG-out, according to the Dunbrack definition. For DDR1, structures were divided into five clusters, with one cluster of size 15 being the closest to the classical DFG-out state.

We subsequently docked type II inhibitors, ponatinib and imatinib, to the most 'classical DFG-out'-like cluster from rMSA AF2 ensembles (indicated as red circles in *Figure 3A and B*), using IFD. Despite the A-loop being folded, the DFG motif in structures from the Abl1 cluster is not strictly DFG-out. As expected, type II ligands fail to dock with those structures, with all IFD poses exhibiting ligand RMSDs above 9 Å (*Figure 3—figure supplement 4A*, Figure 5B). While AF2-based methods can indeed generate decoy structures that deviate from the native state, it remains uncertain whether these decoys correspond to metastable basins. Additionally, it's unclear whether these decoys include structures that can represent the specific metastable states required for the intended types of drug design.

In contrast to Abl1, rMSA AF2 ensemble for DDR1 contains holo-like structure for type II inhibitors. One structure (which we label the 'holo-model', shown as a red circle filled with green in *Figure 3B*), out of the 15 in the DDR1 classical DFG-out cluster docked ponatinib with a remarkably low RMSD

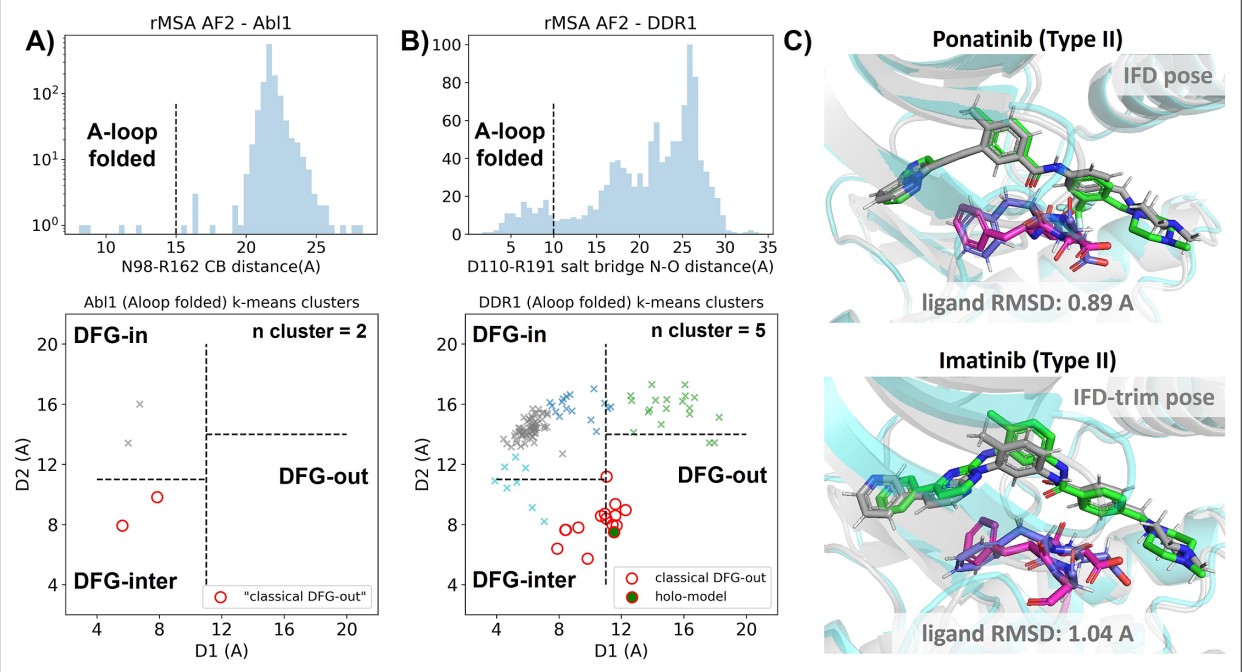

**Figure 3.** Reduced MSA AF2 is capable of generating crystal-like DFG-out state for DDR1 kinase. (**A**) Upper panel: distribution of A-loop location for the reduced multiple sequence alignment (MSA) AlphaFold2 (AF2) structures of Abl1 kinase. 4 out of 1198 structures are A-loop folded. Lower panel: k-means clustering of four A-loop folded Abl1 structures in Dunbrack space, with the number of clusters (n cluster) set to 2. The structures in the cluster closest to the classical DFG-out state (red circles) fail to dock with type II inhibitors using Induced Fit docking (IFD). (**B**) Upper panel: distribution of A-loop location for the reduced MSA AF2 structures of DDR1 kinase. 124 out of 1147 structures are A-loop folded. Lower panel: k-means clustering of 124 A-loop folded DDR1 structures in Dunbrack space, with n cluster set to 5. Among the 15 structures in the cluster closest to the classical DFG-out state (red circles), one structure ('holo-model', highlighted by a red circle filled with green) demonstrates successful docking with type II inhibitors, showcasing a ligand RMSD<2 Å, utilizing IFD or an extended-sampling version of IFD, IFD-trim. (**C**) The docking poses with the lowest ligand RMSD for two type II kinase inhibitors targeting the DDR1 kinase 'holo-model' structure, generated by IFD or IFD-trim. The color code is the same as *Figure 2*.

The online version of this article includes the following figure supplement(s) for figure 3:

**Figure supplement 1.** The plot illustrates the number of gaps in the multiple sequence alignment (MSA) generated by mmseq2 (using ColabFold; *Mirdita et al., 2022*) for different kinases.

**Figure supplement 2.** The AlphaFold2 (AF2) pLDDT rank is plotted against the CA RMSDs from the AF2 structure (the one with the highest pLDDT) for each structure in the reduced multiple sequence alignment (rMSA) AF2 ensemble for Abl1, DDR1, or Src kinase.

**Figure supplement 3.** The AlphaFold2 (AF2) pLDDT rank is plotted against the CA RMSDs from the AF2 structure for each structure in the AF2-cluster ensemble for Abl1, DDR1, or SrcK.

**Figure supplement 4.** Ligand RMSDs are plotted against the docking scores for the Induced Fit docking (IFD) poses of type II inhibitors (ponatinib and imatinib) against AlphaFold2 (AF2) structure (blue) or classical DFG-out structures in reduced multiple sequence alignment (rMSA) AF2 ensembles (red).

**Figure supplement 5.** Docking studies on holo-model from rMSA AF2 for DDR1 kinase.

**Figure supplement 6.** Ligand RMSDs are plotted against the DiffDock confidence scores for the DiffDock poses of type II inhibitors (ponatinib and imatinib) against DDR1 AlphaFold2 (AF2) structure (blue) or the classical DFG-out structures in DDR1 reduced multiple sequence alignment (rMSA) AF2 ensemble (red).

of 0.89 Å using IFD (*Figure 3C*). However, IFD poses from all the other structures exhibit large ligand RMSDs above 6 Å (*Figure 3—figure supplement 4B*). Hence, an enrichment process to select holo-like structures from decoys becomes essential to ensure a practical number of pocket structures for ensemble docking and virtual screening.

We have also docked the 'holo-model' structure with another type II ligand, imatinib. In this case, the steric clashes are more significant, rendering the refinement for the induced fit effect more challenging compared to the ponatinib scenario. We thus introduced an extra trimming step (see Methods section for details) for the DFG-Phe residue which exhibit significant clashes with the holo-imatinib in the 'holo-model' structure (*Figure 3—figure supplement 5B*). The IFD-trim protocol successfully

achieves a minimum ligand RMSD of 1.04 Å for docking poses of the 'holo-model' structure with imatinib (*Figure 3C*, *Figure 3—figure supplement 5C and D*).

## AF2RAVE on DDR1 enriches holo-like classical DFG-out structures in rMSA AF2 decoys

To assess the utility of physics-based methods in selecting holo-like structures from AF2-generated ensembles, we utilized a physics-based protocol, AF2RAVE (*Vani et al., 2023*), to explore the energy landscape of DDR1 kinase. We employed the identical set of collective variables (CVs) as in our previous study (*Vani et al., 2024*) to perform regular space clustering for the DDR1 rMSA AF2 ensemble. Two structures were selected from the clustering centers for each combination of DFG type (in, inter, or out) and A-loop position (folded or extended), resulting in a total of 12 initial structures (*Figure 4—figure supplement 1*). Subsequently, 50 ns unbiased MD simulations were conducted starting from each initial structure. The CVs extracted from all MD trajectories were input into the SPIB model with a linear encoder, to learn the reaction coordinates indicative of the slow motions linked to transitions between metastable states. The learnt SPIB reaction coordinates possess clear physical interpretations, with the first coordinate signifying the A-loop position and the second indicating the DFG type (*Figure 4A and B*). This alignment with physical features is expected as we deliberately selected diverse and representative initial structures for unbiased MD simulations. Additionally, it's noteworthy that these reaction coordinates are transferable across various kinases and can effectively discern different metastable states.

Interestingly, the 15 classical DFG-out structures within the DDR1 rMSA AF2 ensemble are situated in a region of the latent space that were not thoroughly explored by the 12 unbiased MD trajectories. To address this gap, we employed enhanced sampling to sample along the SPIB-approximated reaction coordinates and compute the free energy profile inside the classical DFG-out basin. Considering both the flipping of the DFG motif and the overall motion of the large flexible A-loop, it might be impractical to sample direct back-and-forth transitions between various states using metadynamics. Therefore, we opted for umbrella sampling for its simplicity. The reliability of umbrella sampling hinges on two issues, first whether the latent space adequately represents the conformational space and second, whether there is sufficient overlap between different windows for efficient reweighting. Addressing the first challenge remains an ongoing endeavor in the dimensionality reduction research field, and we anticipate that our SPIB latent space is sufficient enough for our current purpose. The second challenge can be managed through careful setup of umbrella sampling windows and bias strength. Given our current setup, sampling the extensive motion of A-loop relocation remains challenging, showing insufficient overlap between the A-loop folded and extended regions (*Figure 4—figure supplement 2*). Consequently, the quantitative reliability of the absolute $\Delta G$ values between different states is limited, allowing us only to qualitatively assess the relative thermodynamic stability of the DFG-in versus the DFG-out basins. Nevertheless, the qualitative relative stability observed from umbrella sampling aligns with previous studies (*Vani et al., 2024*; *Hanson et al., 2019*; *Figure 4—figure supplement 3*). Furthermore, the local potential of mean force (PMF) surrounding each basin should provide quantitatively reliable insights, given their thorough sampling through umbrella sampling. Reassuringly, when we ranked the classical DFG-out structures within the DDR1 rMSA AF2 ensemble, using the local PMF values in the latent space, the 'holo-model' structure emerged among the top 2 structures with free energy relative to the minimum smaller than 1 kJ/mol, as illustrated in *Figure 4D*.

Additionally, we used DiffDock to conduct docking experiments with type II ligands on the 15 classical DFG-out structures within the DDR1 rMSA AF2 ensemble. Despite poses from most structures surprisingly demonstrating very low ligand RMSD, typically below 2 Å (*Figure 3—figure supplement 6*), they exhibited significant steric clashes and low DiffDock confidence score due to DiffDock's disregard for side-chain configurations and clashes. Notably, upon ranking structures based on their AF2RAVE PMF and examining the corresponding poses with the lowest ligand RMSD, we observed that only the poses derived from the top 2 structures selected by AF2RAVE had DiffDock confidence scores higher than –1.5, surpassing the default threshold of the DiffDock model (*Supplementary file 1*). This implies the general advantage of structures selected by physics-based methods across various docking methods.

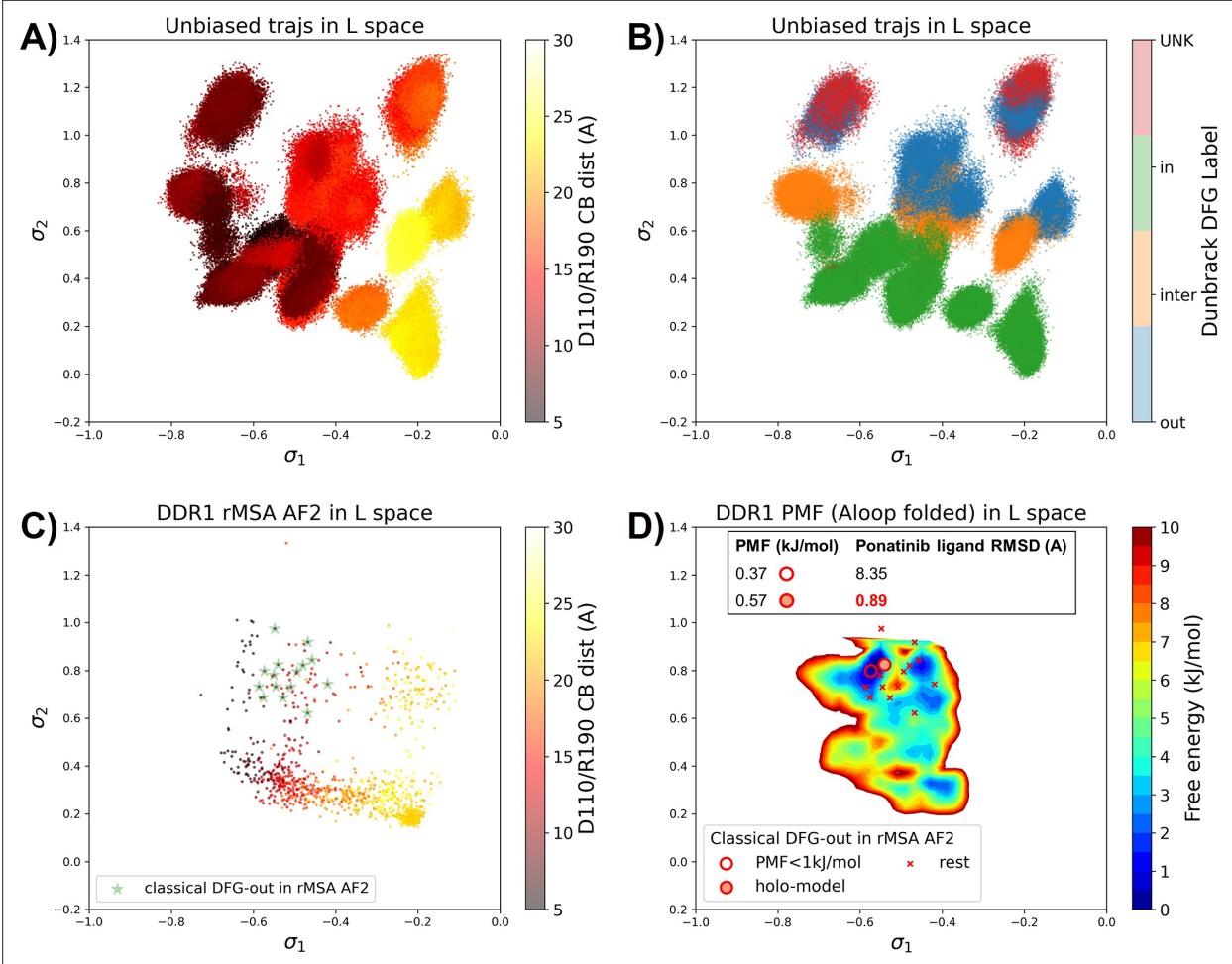

**Figure 4.** Ranking the DDR1 structures using AF2RAVE protocol on the learnt latent space. (**A or B**) The unbiased molecular dynamics (MD) trajectories of DDR1 are projected onto the learnt state predictive information bottleneck model (SPIB) latent space. In plot (**A**), the colors of sample points represent the A-loop location, while in plot (**B**), they depict the Dunbrack DFG state. The first SPIB coordinate, $\sigma_1$, correlates with the A-loop location, and the second SPIB coordinate, $\sigma_2$, correlates with configuration of the DFG motif. (**C**) The reduced multiple sequence alignment (rMSA) AlphaFold2 (AF2) structures of DDR1 are projected onto the latent space. Sample points are color-coded based on the A-loop location. Light green stars highlight the 15 classical DFG-out structures selected based on prior information in *Figure 3*. (**D**) Free energy profile in the A-loop folded region of the latent space, calculated from umbrella sampling simulations. The 15 classical DFG-out structures from rMSA AF2 are shown as red cross and circles (structures with free energy less than 1 kJ/mol). The 'holo-model' structure is emphasized using a red circle filled with red. The embedding table shows the lowest ligand RMSD in IFD poses of the rMSA AF2 structure with ponatinib. The 'holo-model' is among the two structures selected by AF2RAVE (potential of mean force [PMF]<1 kJ/mol).

The online version of this article includes the following figure supplement(s) for figure 4:

**Figure supplement 1.** Reduced MSA AF2 generated DDR1 kinase structures on the Dunbrack space.

**Figure supplement 2.** Umbrella sampling for DDR1 kinase.

**Figure supplement 3.** DDR1 potential of mean force (PMF) calculated with all the umbrella sampling windows.

**Figure supplement 4.** Potential of mean force (PMF) values and Boltzmann ranks of candidate structures fluctuate with the selection of the umbrella sampling windows and the simulation length of umbrella sampling trajectories, demonstrated with the DDR1 system.

**Figure supplement 5.** Free energy profile for DDR1 in the latent space, calculated from unbiased molecular dynamics (MD) simulations.

**Figure supplement 6.** Unbiased simulation coverage study on DDR1 and Abl1 kinase.

In summary, Boltzmann ranking from AF2RAVE effectively distinguishes holo-like structures from other decoys. Beginning with the DDR1 rMSA AF2 ensemble, AF2RAVE substantially enhances the likelihood of identifying the holo-like structure from 1 out of 15 to at least 1 out of 2 when using a PMF cutoff of 1 kJ/mol. We noted that although the PMF values and Boltzmann ranks may fluctuate

with the setup of umbrella sampling, the enrichment effect of holo-like structures remains (*Figure 4—figure supplement 4*). Besides, we applied an alternative protocol to compute the PMF profile within the DDR1 classical DFG-out basin, by running unbiased MD simulations starting from the 15 classical DFG-out decoys in the DDR1 rMSA AF2 ensemble. However, this unbiased protocol poses a risk of failing to sample rare events if barriers exit within the region of interest. In this case, the mini-barriers inside the classical DFG-out basin are low enough to enable efficient sampling using 50 ns unbiased MD simulations. Consequently, the Boltzmann ranks derived from unbiased MD simulations also successfully enriched the single holo-like structure in the DDR1 rMSA AF2 ensemble to the top 5 of the structure set (*Figure 4—figure supplement 5*).

## Transferable learning of holo-like structure for Abl1 and Src kinases with AF2RAVE from DDR1 templates

As mentioned earlier, our current Abl1 rMSA AF2 ensemble lacks any decoy structure in the classical DFG-out state. This is further illustrated in *Figure 5A*, where the Abl1 rMSA AF2 ensemble is projected onto the same latent space learnt during the DDR1 AF2RAVE protocol. Hence, it's necessary to prepare the Abl1 decoy set adopting the classical DFG-out state before we can rank them using Boltzmann weights derived from physics-based methods and conduct further docking for selected holo-like structures.

There are several approaches to generate the Abl1 decoy sets. One can conduct enhanced sampling on the latent space starting from Abl1 rMSA AF2 structures to reach the classical DFG-out basin. Subsequently, MD structures from this basin can serve as templates for asking AF2 to generate crystal-like structures in classical DFG-out state. However, for simplicity, we opted to use the 15 classical DFG-out structures from the DDR1 rMSA AF2 ensemble directly as templates and employed an AF2-based homology modeling protocol, referred to as AF2-template (tAF2 for short, detailed protocol can be found in the Methods section), to generate a decoy set comprising 30 Abl1 structures, as illustrated by the green stars in *Figure 5A*.

Compared to the AF2 and rMSA AF2 structures, the performance of IFD on type II inhibitors shows a significant improvement when using the 30 tAF2 decoy structures of Abl1. The lowest ligand RMSD achieved is 2.74 Å for imatinib and 0.78 Å for ponatinib (*Figure 5B*). However, only four structures (labeled as 'holo-model' structures hereafter) out of the 30 decoys are capable of docking with type II inhibitors with ligand RMSD <3 Å, and all other structures produce IFD poses with ligand RMSD>6 Å (*Figure 5—figure supplement 1*).

We then investigated whether physics-based methods could enrich the 4 'holo-model' structures from the 30 tAF2 decoys to a practical number of holo-like candidates. To explore the local classical DFG-out basin, we conducted 50 ns unbiased MD simulations starting from each tAF2 decoy structure and combined the trajectories to obtain the final Boltzmann distribution. Remarkably, the PMF for the Abl1 classical DFG-out basin calculated from the unbiased protocol (*Figure 5C*) showed the enrichment of all 4 'holo-model' structures among the top 8 tAF2 structures with PMF value <1 kJ/mol. The PMF values of the top 8 tAF2 structures and the lowest ligand RMSD from the corresponding structures are presented in *Figure 5D*. We also employed umbrella sampling for the Abl1 kinase by transferring setups and initial structures using the AF2-template from DDR1 umbrella sampling. However, we noted a significant occurrence of αC helix breakage in the Abl1 umbrella sampling trajectories compared to DDR1 (*Figure 5—figure supplement 6*). After excluding windows with broken αC helix, the Abl1 PMF derived from umbrella sampling ultimately enriched all 4 'holo-model' structures to the top 6 among the 30 tAF2 structures (*Figure 5—figure supplement 7*). In Supplementary Information, we also provide results for Src kinase which are in the same quality as for Abl1 kinase.

## Discussion

Through our retrospective analysis, we have thus demonstrated that the default AF2 models are ineffective for docking ligands targeting metastable protein kinase conformations. While AF2-based methods can be coaxed into generating diverse structures, they still struggle to produce reliable accuracy decoys for metastable conformations since the AF2 ensembles do not follow Boltzmann distribution. This failure is evident in the inability to generate an Abl1 AF2 ensemble containing holo-like structures for type II inhibitors. To further investigate whether this limitation is common among

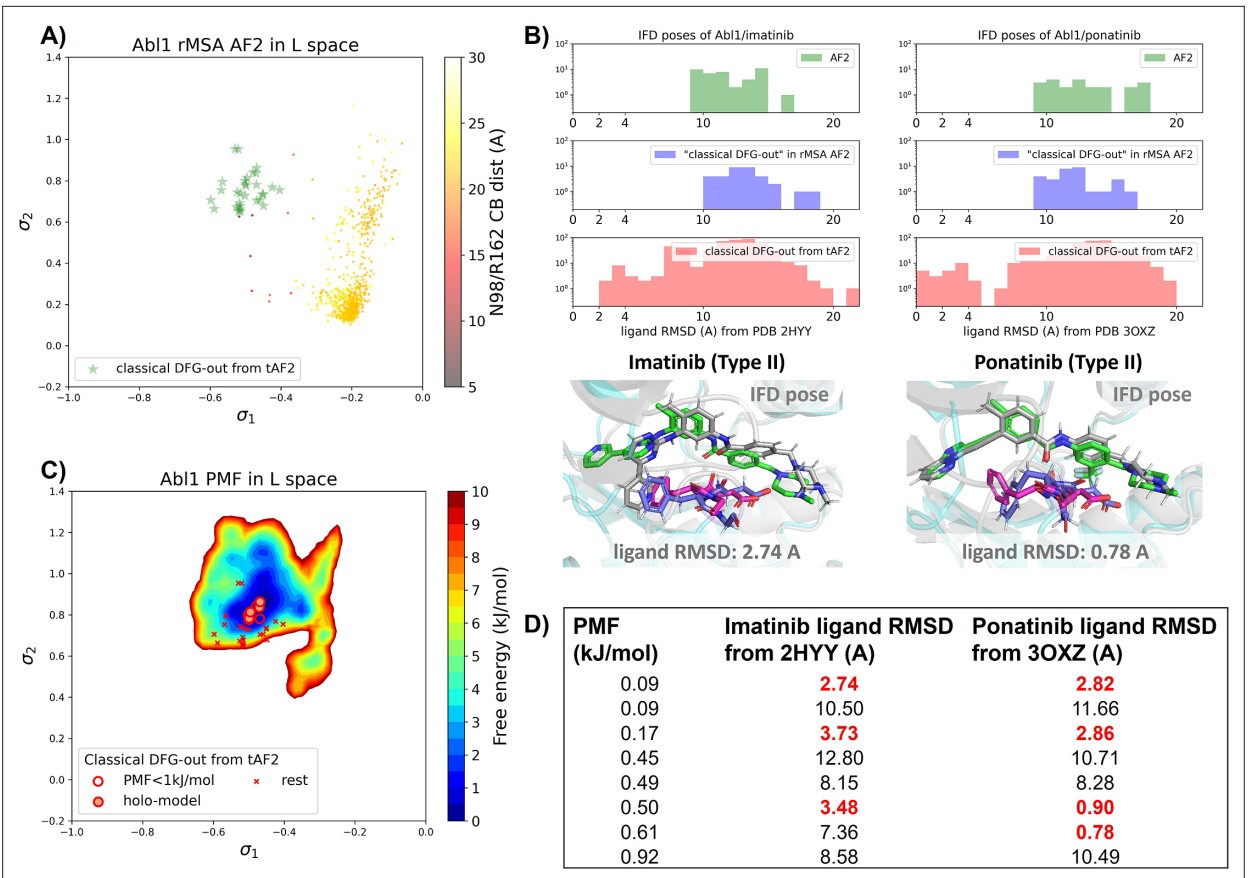

**Figure 5.** Transferrable learning of holo-like structure for Abl1 kinase. (**A**) The reduced multiple sequence alignment (MSA) AlphaFold2 (AF2) structures of Abl1 are projected onto the latent space. Sample points are color-coded based on the A-loop location. Light green stars highlight the 30 AF2-template Abl1 structures modeled from the 15 DDR1 classical DFG-out structures. (**B**) Upper panel: the distribution of ligand RMSD for the Induced Fit docking (IFD) poses of Abl1 structures and two type II ligands. Lower panel: IFD poses with the lowest ligand RMSD for Abl1 AF2-template structures and two type II ligands. The color code is the same as *Figure 2*. (**C**) Free energy profile in the latent space, calculated from unbiased molecular dynamics (MD) simulations. The 30 Abl1 classical DFG-out structures from AF2-template are shown as red cross and circles (structures with free energy less than 1 kJ/mol). The 'holo-model' structures are emphasized using red circles filled with red. (**D**) The table shows the lowest ligand RMSD in IFD poses of the AF2-template structures with two type II inhibitors. All the four 'holo-models' are among the eight structures selected by AF2RAVE (potential of mean force [PMF]<1 kJ/mol).

The online version of this article includes the following figure supplement(s) for figure 5:

**Figure supplement 1.** Ligand RMSDs are plotted against the docking scores for the Induced Fit docking (IFD) poses of type II inhibitors (ponatinib and imatinib) against Abl1 tAF2 structures.

**Figure supplement 2.** The projection of A-loop folded structures from the reduced multiple sequence alignment (rMSA) AlphaFold2 (AF2) ensemble or the AF2-cluster ensemble (cAF2) on the AF2RAVE potential of mean force (PMF) for Abl1 or DDR1.

**Figure supplement 3.** Structural visualization of tAF2 generated classical DFG-out Src kinase.

**Figure supplement 4.** Comparison between performance of rMSA AF2 and AF2-cluster on Src kinase.

**Figure supplement 5.** Ligand RMSDs are plotted against the docking scores for the Induced Fit docking (IFD)/IFD-trim docking poses of type II inhibitors (ponatinib and imatinib) against the SrcK tAF2 structure.

**Figure supplement 6.** Accounting for broken αC helix during umbrella sampling of DDR1 and Abl1 kinase.

**Figure supplement 7.** Abl1 potential of mean force (PMF) calculated from umbrella sampling after discarding windows with αC helix broken.

AF2-based methods, including the rMSA AF2 method we employed earlier, we tested another AF2-based approach, AF2-cluster (*Wayment-Steele et al., 2024*). The AF2-cluster ensemble of Abl1 comprises more A-loop folded structures (21 out of 197) compared to our Abl1 rMSA AF2 ensemble (4 out of 1198). However, similar to rMSA AF2, there are still no decoys in the classical DFG-out state, and all A-loop folded structures are located far from the PMF basin of the classical DFG-out state in the

**Table 1.** Comparing the Induced Fit docking (IFD) performance of various structure generation methods for docking type II kinase inhibitors.

| Source-protein (# of structures) | Lowest imatinib ligand RMSD (Å) | Lowest ponatinib ligand RMSD (Å) | Ratio of structs. w. ligand RMSD <3 Å |
|---|---|---|---|
| AF2-Abl1 (1) | 9.22 | 9.4 | 0/1 |
| AF2-DDR1 (1) | 9.24 | 9.33 | 0/1 |
| rMSA AF2-Abl1 (2) | 10.14 | 9.11 | 0/2 |
| rMSA AF2-DDR1 (15) | 1.04* | 0.89 | 1/15 |
| tAF2-Abl1 (30) | 2.74 | 0.78 | 4/30 |
| AF2RAVE-DDR1 (2) | 1.04* | 0.89 | 1/2 |
| AF2RAVE-Abl1 (8) | 2.74 | 0.78 | 4/8 |

*Result from IFD-trim.

latent space (*Figure 5—figure supplement 2C*). This indicates that AF2-cluster, like rMSA AF2, also fails to generate Abl1 metastable states effectively. Interestingly, we observed comparable structural diversity in sampling the A-loop folded configurations within the AF2-cluster ensembles for DDR1 and Abl1 (*Figure 5—figure supplement 2C and D*). While for the rMSA AF2 method, the promiscuous kinase DDR1 ensemble exhibits superior structural diversity compared to the Abl1 kinase. This enhanced diversity leads to the identification of one dockable structure for type II inhibitors among decoys in DDR1 rMSA AF2 ensemble. Through the application of a homology modeling method, AF2-template, we demonstrated that the classical DFG-out decoys in the DDR1 rMSA AF2 ensemble can be transferred to Abl1 kinase. Furthermore, we tested an additional kinase, Src, for which non-native state decoys are reported to be even more challenging to produce using AF2 subsampling methods than Abl1 kinase (*Monteiro da Silva et al., 2024*). We have also verified that rMSA AF2 and AF2-cluster struggle in producing distinct decoys from native structure of Src kinase (*Figure 3—figure supplement 2*, *Figure 3—figure supplement 3*). Besides, neither the rMSA AF2 nor the AF2-cluster ensembles for Src kinase adequately sample the classical DFG-out basin in the latent space (*Figure 5—figure supplement 4*). Remarkably, AF2-template, as a homology modeling method, can easily produce the classical DFG-out structure of Src kinase, using the top 2 AF2RAVE-picked DDR1 classical DFG-out structures as templates (*Figure 5—figure supplement 3*). The IFD poses from the tAF2 structure of Src kinase shown a minimal ligand RMSD of 2.82 Å with imatinib (*Figure 5—figure supplement 5*).

With the rapid expansion of chemical space, virtual screening on libraries containing billions of diverse molecules becomes enticing for novel drug discovery (*Lyu et al., 2019*). Therefore, the enrichment of candidate holo-like structures emerges as a necessary step, offering significant benefits in terms of computational efficiency and feasibility. As summarized in *Table 1*, unlike the non-dockable AF2 structures for type II inhibitors, the diverse rMSA AF2 ensemble (referred to as rAF2 in *Table 1* for brevity) shows potential in generating holo-like structures within a large set of decoys. However, it's only upon AF2RAVE ranking and selection that the ratio of holo-like structures in selected structure set increases to a plausible value of 50%, facilitating further virtual screening on computational models of protein pockets.

## Conclusion

AF2 has arguably revolutionized protein structure prediction, but it remains to be constructively demonstrated if it can be reliably used for drug discovery purposes, especially involving non-native protein conformations. In this work we have demonstrated through retrospective studies on kinase inhibitors that a combination of AF2, statistical mechanics-based enhanced sampling, and IFD can be deployed for such calculations. Specifically, we have utilized the AF2RAVE protocol by inputting the sequences of the DDR1 kinases, along with two additional pieces of prior information: a pairwise distance cutoff for evaluating A-loop positions and the Dunbrack definition of DFG type. We then employed Glide IFD to assess our AF2RAVE-generated computational models for type II kinase inhibitor binding pockets. This AF2RAVE-Glide workflow yielded holo-like structure candidates with a 50%

successful docking rate for known type II inhibitors. Notably, the holo-like structures in metastable state and the latent space constructed from AF2RAVE of DDR1 are transferable to other kinases. This includes the challenging cases (*Monteiro da Silva et al., 2024*) of Abl1 and Src kinases, wherein we showed that SPIB and sampling performed for DDR1 allowed generating classical DFG-out structures for both Abl1 and Src kinases. This severely reduces the computational cost for retraining SPIB to learn low-dimensional latent space for different kinases.

This demonstration of AF2RAVE-Glide on kinase inhibitors shows its promising application for discovering drugs targeting general proteins in addition to kinases, such as G-protein-coupled receptors, which are the targets for over one-third of Food and Drug Administration (FDA)-approved drugs (*Hauser et al., 2018*). For the design of novel drugs targeting general proteins, developing a protocol that does not require prior information about the system is left for future exploration. Besides, in this study we only investigate the classical DFG-out metastable state for kinases. For a comprehensive protocol, all top-ranked metastable states identified by AF2RAVE should be explored in subsequent docking experiments. Integration of algorithms capable of predicting ligand binding sites on protein surfaces, such as the Graph Attention Site Prediction (GrASP) model (*Smith et al., 2024*), is then essential before utilizing AF2RAVE-selected structures in docking, thus expanding the workflow to AF2RAVE-GrASP-Glide. Additionally, the inclusion of FEP calculations for front-runner ligands to evaluate the actual ligand binding affinity can further enhance this workflow.

The integration of AF2-based and physics-based methods presents a promising approach toward the development of a mature workflow for computer-aided drug design. AF2-based methods are capable of producing ensembles with structural diversity, which aids physics-based methods in better sampling and exploring the energy landscape of proteins. Additionally, AF2-generated structures can serve as crystal-like decoys, free from distortion that may occur in biased simulations. Physics-based methods play a crucial role in accurately assigning Boltzmann weights and ranking decoy structures to guide the enrichment of holo-like structures, essential for virtual screening on large libraries. This collaborative approach leverages the strengths of both methodologies, leading to enhanced efficiency and efficacy in drug discovery efforts.

We conclude this manuscript by making a final comment on the horizons opened by generative AI methods, including those involving diffusion models (*Abramson et al., 2024*; *Wang et al., 2022*) and any such future frameworks. These approaches make it possible to easily hypothesize regions of the conformational space underlying arbitrarily complex molecules of life, that then could serve as a starting point to launch more careful investigations. However, these predictions without careful in situ or a posteriori testing through advanced simulations or experiments are only predictions and could just be hallucinations. Having the capability to quickly generate numerous - thousands or more - such structural hypotheses is what made AF2 so crucial to this current work. We believe that a deep integration of the hypothesis creation possibilities of generative AI with careful MD (*Herron et al., 2023*; *Zheng et al., 2024*) experiments or other forms of rapid testing is one way these methods will truly facilitate new and reliable scientific discoveries.

## Methods
### Summary of systems tested and tools used

In this work we tested AF2RAVE-Glide protocol on active and inactive conformations of DDR1, Abl1, and Src kinases against type I inhibitors VX-680 and dasatinib, and type II inhibitors imatinib and ponatinib. The top-ranked structures obtained from AF2RAVE-Glide were compared against publicly available crystal structures deposited in the Protein Data Bank (PDB), with PDB codes provided in the main text.

To generate diverse structural ensembles for DDR1, Abl1, and Src kinases, we primarily used the rMSA AF2 method. For comparison, we also produced AF2-cluster ensembles for these three kinases.

For MD simulations, all the systems in this paper were parameterized with the Amber99SB*-ILDN force field (*Case et al., 2005*; *Lindorff-Larsen et al., 2010*) with the TIP3P water model (*Jorgensen et al., 1983*) and neutralized with Na$^+$ ions or Cl$^-$ ions. The simulations are performed at 300 K with the LangevinMiddleIntegrator (*Zhang et al., 2019*) in OpenMM (*Eastman et al., 2017*) with the step size of 2 fs. Particle mesh Ewald (*Darden et al., 1993*) is used for calculating electrostatics and the lengths of bonds to hydrogen atoms are constrained using LINCS (*Hess et al., 1997*) throughout all

simulations. Before performing MD simulations for analysis, energy minimization is conducted for all initial structures, followed by equilibrium runs under NVT and NPT for 500 ps and 1 ns respectively.

To account for the induced fit effect, IFD from the Schrödinger suite is the primary docking method used in this work. For comparison, we also tested the Glide XP docking from the Schrödinger suite and DiffDock to dock the two known type II inhibitors (imatinib and ponatinib) against DDR1 AF2 structure and the 15 classical DFG-out structures in the DDR1 rMSA ensemble.

Code and data for this paper are available in https://github.com/tiwarylab/AF2RAVE_Glide-kinase.

## Generation of rMSA AF2 ensemble, AF2-cluster ensemble, and AF2-template structures

In this work, ColabFold (*Mirdita et al., 2022*) was employed to generate all the AF2-based ensembles and structures. The MSAs were produced using mmseq2. For rMSA AF2 ensembles, the MSA depth was reduced to either 16 or 32 for each kinase. For each depth, 128 random seeds were utilized, and each seed produced 5 structural models via ColabFold, resulting in a total of 1280 rMSA AF2 structures per kinase. The structure with the highest pLDDT score among all 1280 was identified as the AF2 structure (native structure). Subsequently, any unphysical structures with an RMSD greater than 7 Å from the AF2 structure were discarded, culminating in the final rMSA AF2 ensembles.

AF2-cluster for DDR1, Abl1, and SrcK were run with default setups as provided in the ColabFold notebook in the original paper of AF2-cluster (*Wayment-Steele et al., 2024*).

The AF2-based homology modeling protocol, AF2-template (tAF2) method, is implemented using ColabFold. For a given query sequence, tAF2 structures are generated by ColabFold upon uploading desired template structure and deactivating the Evoformer module. For each template, five AF2-template models are generated, and the last three structures exhibiting lower pLDDT will be discarded. The AF2-template (tAF2) method is employed here to transfer structure sets between homologous systems. To generate tAF2 structures for Abl1 in the classical DFG-out state, each of the 15 classical DFG-out structures from the DDR1 rMSA AF2 ensemble is used as a template, resulting in 30 tAF2 Abl1 structures in total. For Src kinase, a single representative tAF2 structure is generated using the 'holo-model' DDR1 structure as the template.

## AF2RAVE protocol

### Regular space clustering on the rMSA AF2 ensemble

We used the same 14 CVs for regular space clustering as in the previous AF2RAVE work on kinases (*Vani et al., 2024*). These CVs are pairwise distances, selected to describe the kinase conformations around the ATP-binding pocket and the A-loop.

### Unbiased MD and SPIB

50 ns unbiased MD simulation was run for each AF2RAVE initial structure of DDR1 from *Figure 4—figure supplement 1*. The standard deviations of the 14 CVs are calculated after concatenating all the 12 unbiased MD trajectories. 8 CVs with standard deviations larger than 0.25 of the maximum standard deviation remain as the input features of the SPIB model. We conducted a parameter screening of the SPIB time lag, ranging from 1 ns to 40 ns with intervals of 1 ns. Eventually, we selected a time lag of 16 ns based on the performance of SPIB coordinates in representing physical features, including DFG type and A-loop position.

### PMF calculations from umbrella sampling

2D umbrella sampling is conducted along the two learnt SPIB coordinates, employing 11×11 windows. The bias potential equilibrium points are uniformly distributed in the SPIB latent space, with $\sigma_1$ ranging from –0.8 to –0.1 and $\sigma_2$ ranging from 0 to 0.8. The strength of the bias potential is set to 1000kJ/mol/nm$^2$. Each window originates from the structure closest to the window's equilibrium point in Euclidean distance within the latent space among the 12 AF2RAVE initial structures and lasts for 100 ns (with the first 10 ns discarded in PMF calculation). For Abl1 kinase, we noticed a substantial proportion of the αC helix breaking in the umbrella sampling trajectories (*Figure 5—figure supplement 6*). This finding aligns with earlier enhanced sampling investigations on DDR1 using metadynamics (*Vani et al., 2024*), where the authors imposed restraints to prevent αC helix breakage. In our study, we opted to exclude

all umbrella sampling windows where the ratio of broken αC helix exceeded 20% for the Abl1 PMF calculation (*Figure 5—figure supplement 7*). The WHAM algorithm is applied to bin and reweight the biased trajectories and compute the final PMF.

## PMF calculations from unbiased MD

50 ns unbiased MD simulation was run starting from each structure in the 15 classical DFG-out structures in DDR1 rMSA AF2 ensemble. Upon discarding first 10 ns, all the unbiased trajectories are simply concatenated to calculate the Boltzmann distribution and PMF for each bin around the classical DFG-out basin in the latent space. For Abl1, unbiased simulations start from 30 AF2-template structures to calculate the PMF around the classical DFG-out basin.

## Boltzmann ranks assignment for structures in AF2-based ensembles

After calculating the PMF value for each bin in the latent space, we projected AF2-generated structures into latent space. We then directly assign the PMF values of the corresponding bins to these AF2-generated structures.

We must acknowledge the limitations of the way we assigned PMF values to AF-generated candidate holo structures. First of all, the free energy profiles are derived from MD simulations, and the PMF values directly correspond to the MD structures. Here, we assumed that the latent space adequately represents the conformational changes of protein pocket within specific metastable states. Additionally, while the enrichment of holo structures in the top Boltzmann-ranked structures persists, the absolute PMF values and Boltzmann ranks of proper holo structures may fluctuate with the umbrella sampling setups, as depicted in *Figure 4—figure supplement 4*. Theoretically, the number of umbrella sampling windows and the simulation length should be sufficiently large for PMF convergence. However, there is always a trade-off between PMF accuracy and computational costs, so we opted to stick with the current setups.

## Docking details

All the input structure for our docking experiments were first relaxed in solution with an MD energy minimization step. This work investigates two type I inhibitors (VX-680 and dasatinib) and two type II inhibitors (imatinib and ponatinib) by docking. For Glide XP docking or IFD, we used Ligprep in Maestro to prepare the ligand inputs from SMILES files. For DiffDock, the ligand inputs were directly provided as SMILES files.

### Glide XP docking

Glide XP docking experiments in this work were run with default setups in the Maestro software. Glide XP docking was performed for all four ligands on the AF2 structures of DDR1 or Abl1, as well as on the 15 classical DFG-out conformations of DDR1 in the rMSA AF2 ensemble.

### Induced Fit docking

For IFD, we used the Glide XP for initial docking, followed by Prime relaxation and final Glide XP docking. Parameters remain default in Maestro IFD. IFD was performed only for the type II ligands on the AF2 structures of DDR1 or Abl1, on the classical DFG-out conformations of DDR1 or Abl1 in the rMSA AF2 ensembles, as well as tAF2 structures of Abl1 and SrcK.

Given ponatinib's backbone features, notably its lengthy and slender carbon-carbon triple bond, it exhibits reduced sensitivity to steric clashes, resulting in successful docking with the DDR1 'holo-model' structure at a ligand RMSD of 0.89 Å. Conversely, the 'holo-model' structure struggles to accurately dock with imatinib using IFD (*Figure 3—figure supplement 4B*). We then employed an extended-sampling IFD approach by initially trimming the DFG-Phe residue from the 'holo-model' structure. The trimmed residue is temporarily mutated to alanine during initial docking and later restored in subsequent Prime relaxation and final docking steps. Here, in the 'holo-model' structure, we manually chose the DFG-Phe residue which is significantly hindered by holo-imatinib (*Figure 3— figure supplement 5B*). For generic systems lacking groundtruth information, broader screening of single residue trimming for the protein pocket may be necessary for this extended-sampling IFD

method. In essence, it's a trade-off between the quality of holo-like structure to dock with and the accuracy/complexity of the docking method.

## DiffDock performance on DDR1 classical DFG-out in rMSA AF2 ensemble

DiffDock docking experiments in this work were run in the webserver with default setups in the version before March 8, 2024 (https://huggingface.co/spaces/simonduerr/diffdock; *Corso et al., 2022*). Diff-Dock was performed only for type II ligands on the AF2 structures of DDR1, as well as on the 15 classical DFG-out conformations of DDR1 in the rMSA AF2 ensemble.

## Acknowledgements

Research in this publication was supported by the National Institute of General Medical Sciences of the National Institutes of Health under Award Number R35GM142719. The content is solely the responsibility of the authors and does not represent the official views of the National Institutes of Health. We thank UMD HPC's Zaratan and NSF ACCESS (project CHE180027P) for computational resources. AA was supported by NCI-UMD Partnership for Integrative Cancer Research. PT is an investigator at the University of Maryland-Institute for Health Computing, which is supported by funding from Montgomery County, Maryland and The University of Maryland Strategic Partnership: MPowering the State, a formal collaboration between the University of Maryland, College Park, and the University of Maryland, Baltimore. We thank UMD HPC's Zaratan and NSF ACCESS (project CHE180027P) for computational resources. We thank Bodhi Vani, Dedi Wang, Zachary Smith, and Anjali Verma for helpful discussions.

## Additional information

### Competing interests

Pratyush Tiwary: P.T. is a consultant to Schrodinger, Inc and is on their Scientific Advisory Board. The other authors declare that no competing interests exist.

### Funding

| Funder | Grant reference number | Author |
| --- | --- | --- |
| National Institute of General Medical Sciences | R35GM142719 | Pratyush Tiwary |

The funders had no role in study design, data collection and interpretation, or the decision to submit the work for publication.

### Author contributions

Xinyu Gu, Conceptualization, Resources, Data curation, Software, Formal analysis, Supervision, Validation, Investigation, Visualization, Methodology, Writing - original draft, Project administration, Writing – review and editing; Akashnathan Aranganathan, Conceptualization, Resources, Data curation, Software, Formal analysis, Validation, Investigation, Visualization, Methodology, Writing - original draft, Writing – review and editing; Pratyush Tiwary, Conceptualization, Formal analysis, Supervision, Funding acquisition, Project administration, Writing – review and editing

### Author ORCIDs

Akashnathan Aranganathan ⓘ http://orcid.org/0000-0001-7938-2141
Pratyush Tiwary ⓘ https://orcid.org/0000-0002-2412-6922

Reviewer #1 (Public Review): https://doi.org/10.7554/eLife.99702.3.sa1
Reviewer #2 (Public Review): https://doi.org/10.7554/eLife.99702.3.sa2
Reviewer #3 (Public Review): https://doi.org/10.7554/eLife.99702.3.sa3
Author response https://doi.org/10.7554/eLife.99702.3.sa4

## Additional files

### Supplementary files

• MDAR checklist

• Supplementary file 1. Comparison between AF2RAVE ranks and DiffDock confidence scores. Confidence score for the DiffDock pose aligns with AF2RAVE potential of mean force (PMF) values. The DiffDock confidence score of the pose with the lowest ligand RMSD (marked in red/bold) from each classical DFG-out structure in DDR1 reduced multiple sequence alignment (rMSA) AlphaFold2 (AF2) ensemble is compared with the AF2RAVE PMF value for corresponding structure (marked in red/bold).

### Data availability

Code and data for this paper are available in https://github.com/tiwarylab/AF2RAVE_Glide-kinase (copy archived at *Gu, 2024*).

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
