## [Editor Report · eLife assessment]

This **important** study demonstrates that combining AlphaFold2 with the author's sampling method AF2-RAVE improves protein-ligand docking for three protein kinases and their inhibitors. The evidence is **compelling** and the results will be of interest to researchers who work on computer-aided drug design.

---

## [Referee Report · Reviewer #1 (Public Review)]

The development of effective computational methods for protein-ligand binding remains an outstanding challenge to the field of drug design. This impressive computational study combines a variety of structure prediction (AlphaFold2) and sampling (RAVE) tools to generate holo-like protein structures of three kinases (DDR1, Abl1, and Src kinases) for binding to type I and type II inhibitors. Of central importance to the work is the conformational state of the Asp-Phy-Gly "DFG motif" where the Asp points inward (DFG-in) in the active state and outward (DFG-out) in the inactive state. The kinases bind to type I or type II inhibitors when in the DFG-in or DFG-out states, respectively.

It is noted that while AlphaFold2 can be effective in generating ligand-free apo protein structures, it is ineffective at generating holo structures appropriate for ligand binding. Starting from the native apo structure, structural fluctuations are necessary to access holo-like structures appropriate for ligand-binding. A variety of methods, including reduced multiple sequence alignment (rMSA), AF2-cluster, and AlphaFlow may be used to create decoy structures. However, those methods can be limited in the diversity of structures generated and lack a physics-based analysis of Boltzmann weight critical to their relative evaluation.

To address this need, the authors combine AlphaFold2 with the Reweighted Autoencoded Variational Bayes for Enhanced Sampling (RAVE) method, to explore metastable states and create a Boltzmann ranking. With that variety of structures in hand, grid-based docking methods Glide and Induced-Fit Docking (IFD) were used to generate protein-ligand (kinase-inhibitor) complexes.

The authors demonstrate that using AlphaFold2 alone, there is a failure to generate DFG-out structures needed for binding to type II inhibitors. By applying the AlphaFold2 with rMSA followed by RAVE (using short MD trajectories, SPIB-based collective variable analysis, and enhanced sampling using umbrella sampling), metastable DFG-out structures with Boltzmann weighting are generated enabling protein-ligand binding. Moreover, the authors found that the successful sampling of DFG-out states for one kinase (DDR1) could be used to model similar states for other proteins (Abl1 and Src kinase). The AF2RAVE approach is shown to result in a set of holo-like protein structures with a 50% rate of docking type II inhibitors.

Overall, this is excellent work and a valuable contribution to the field that demonstrates the strengths and weaknesses of state-of-the-art computational methods for protein-ligand binding. The authors also suggest promising directions for future study, noting that potential enhancements in the workflow may result from the use of binding site prediction models and free energy perturbation calculations.

---

## [Referee Report · Reviewer #2 (Public Review)]

This manuscript explores the utility of AlphaFold2 (AF2) and the author's own AF2-RAVE method for drug discovery. As has been observed elsewhere, the predictive power of docking against AF2 structures is quite limited, particularly for proteins like kinases that have non-trivial conformational dynamics. However, using enhanced sampling methods like RAVE to explore beyond AF2 starting structures leads to a significant improvement.

Comments on revised version:

I'm happy with the changes made.

---

## [Referee Report · Reviewer #3 (Public Review)]

In this manuscript, the authors aim to enhance AlphaFold2 for protein conformation-selective drug discovery through the integration of AlphaFold2 and physics-based methods, focusing on improving the accuracy of predicting protein structures ensemble and small molecule binding of metastable protein conformations to facilitate targeted drug design.

The major strength of the paper lies in the methodology, which includes the innovative integration of AlphaFold2 with all-atom enhanced sampling molecular dynamics and induced fit docking to produce protein ensembles with structural diversity. Moreover, the generated structures can be used as reliable crystal-like decoys to enrich metastable conformations of holo-like structures. The authors demonstrate the effectiveness of the proposed approach in producing metastable structures of three different protein kinases and perform docking with their type I and II inhibitors. The paper provides strong evidence supporting the potential impact of this technology in drug discovery. However, limitations may exist in the generalizability of the approach across other structures, especially complex structures such as protein-protein or DNA-protein complexes.

The authors largely achieved their aims by demonstrating that the AF2RAVE-Glide workflow can generate holo-like structure candidates with a 50% successful docking rate for known type II inhibitors. This work is likely to have a significant impact on the field by offering a more precise and efficient method for predicting protein structure ensemble, which is essential for designing targeted drugs. The utility of the integrated AF2RAVE-Glide approach may streamline the drug discovery process, potentially leading to the development of more effective and specific medications for various diseases.

Comments on revised version:

The revised manuscript looks great to me. I have no further comments.

---

## [Author Response]

The following is the authors’ response to the original reviews.

**Public Reviews:**

**Reviewer #1 (Public Review):**
The development of effective computational methods for protein-ligand binding remains an outstanding challenge to the field of drug design. This impressive computational study combines a variety of structure prediction (AlphaFold2) and sampling (RAVE) tools to generate holo-like protein structures of three kinases (DDR1, Abl1, and Src kinases) for binding to type I and type II inhibitors. Of central importance to the work is the conformational state of the Asp-Phy-Gly "DFG motif" where the Asp points inward (DFG-in) in the active state and outward (DFG-out) in the inactive state. The kinases bind to type I or type II inhibitors when in the DFG-in or DFG-out states, respectively.It is noted that while AlphaFold2 can be effective in generating ligand-free apo protein structures, it is ineffective at generating holo-structures appropriate for ligand binding. Starting from the native apo structure, structural fluctuations are necessary to access holo-like structures appropriate for ligand binding. A variety of methods, including reduced multiple sequence alignment (rMSA), AF2-cluster, and AlphaFlow may be used to create decoy structures. However, those methods can be limited in the diversity of structures generated and lack a physics-based analysis of Boltzmann weight critical to their relative evaluation.To address this need, the authors combine AlphaFold2 with the Reweighted Autoencoded Variational Bayes for Enhanced Sampling (RAVE) method, to explore metastable states and create a Boltzmann ranking. With that variety of structures in hand, grid-based docking methods Glide and Induced-Fit Docking (IFD) were used to generate protein-ligand (kinase-inhibitor) complexes.The authors demonstrate that using AlphaFold2 alone, there is a failure to generate DFG-out structures needed for binding to type II inhibitors. By applying the AlphaFold2 with rMSA followed by RAVE (using short MD trajectories, SPIB-based collective variable analysis, and enhanced sampling using umbrella sampling), metastable DFG-out structures with Boltzmann weighting are generated enabling protein-ligand binding. Moreover, the authors found that the successful sampling of DFG-out states for one kinase (DDR1) could be used to model similar states for other proteins (Abl1 and Src kinase). The AF2RAVE approach is shown to result in a set of holo-like protein structures with a 50% rate of docking type II inhibitors.Overall, this is excellent work and a valuable contribution to the field that demonstrates the strengths and weaknesses of state-of-the-art computational methods for protein-ligand binding. The authors also suggest promising directions for future study, noting that potential enhancements in the workflow may result from the use of binding site prediction models and free energy perturbation calculations.
**Reviewer #2 (Public Review):**
Summary:This manuscript explores the utility of AlphaFold2 (AF2) and the author's own AF2-RAVE method for drug discovery. As has been observed elsewhere, the predictive power of docking against AF2 structures is quite limited, particularly for proteins like kinases that have non-trivial conformational dynamics. However, using enhanced sampling methods like RAVE to explore beyond AF2 starting structures leads to a significant improvement.Strengths:This is a nice demonstration of the utility of the authors' previously published RAVE method.Weaknesses:My only concern is the authors' discussion of induced fit. I'm quite confident the structures discussed are present in the absence of ligand binding, consistent with conformational selection. It seems the author's own data also argues for an important role in conformational selection. It would be nice to acknowledge this instead of going along with the common practice in drug discovery of attributing any conformational changes to induced fit without thoughtful consideration of conformational selection.

The reviewer is correct. We aim to highlight the significant role of conformational selection. To clarify this, we have expanded the discussion on conformational selection in the introduction.

**Reviewer #3 (Public Review):**
In this manuscript, the authors aim to enhance AlphaFold2 for protein conformation-selective drug discovery through the integration of AlphaFold2 and physics-based methods, focusing on improving the accuracy of predicting protein structures ensemble and small molecule binding of metastable protein conformations to facilitate targeted drug design.The major strength of the paper lies in the methodology, which includes the innovative integration of AlphaFold2 with all-atom enhanced sampling molecular dynamics and induced fit docking to produce protein ensembles with structural diversity. Moreover, the generated structures can be used as reliable crystal-like decoys to enrich metastable conformations of holo-like structures. The authors demonstrate the effectiveness of the proposed approach in producing metastable structures of three different protein kinases and perform docking with their type I and II inhibitors. The paper provides strong evidence supporting the potential impact of this technology in drug discovery. However, limitations may exist in the generalizability of the approach across other structures, especially complex structures such as protein-protein or DNA-protein complexes.

Proteins undergo thermodynamic fluctuations and can occasionally reach metastable configurations. It can be assumed that other biomolecules, such as proteins and DNA, stabilize these metastable states when forming protein-protein or protein-DNA complexes. Since our method has the potential to identify these metastable states, it shows promise for designing drugs targeting proteins in allosteric configurations induced by other biomolecules.

The authors largely achieved their aims by demonstrating that the AF2RAVE-Glide workflow can generate holo-like structure candidates with a 50% successful docking rate for known type II inhibitors. This work is likely to have a significant impact on the field by offering a more precise and efficient method for predicting protein structure ensemble, which is essential for designing targeted drugs. The utility of the integrated AF2RAVE-Glide approach may streamline the drug discovery process, potentially leading to the development of more effective and specific medications for various diseases.
**Recommendations for the authors:**

**Reviewer #1 (Recommendations For The Authors):**
Suggestions(1) The computational protocol is found to be insufficient to generate precise values of the relative free energies between structures generated. The authors note in the Conclusion that an enhancement in the workflow might result from the addition of free energy calculations. Can the authors comment on the prospects for generating more accurate estimates of the free energy that might be used to qualitatively evaluate poses and the free energy landscape surrounding putative metastable states? What are the principal challenges and what might help overcome them? What would the most effective computational protocol be?

More accurate estimates of the free energy can theoretically be achieved by increasing the number of umbrella sampling windows and extending the simulation length until the PMF converges. However, there is always a trade-off between PMF accuracy and computational costs, so we have chosen to stick with the current setup. Metadynamics is another method to obtain a more accurate free energy profile, which we have used in previous versions of AlphaFold2-RAVE, but for the specific systems we investigated, it had issues in achieving back and forth movement given the high entropic nature of the activation loop. Research in enhanced sampling methods and dimensionality reduction techniques for reaction coordinates is continually evolving and will play a critical role in alleviating this problem.

(2) I was surprised that there was not more correlation of a funnel-like shape in Figures S16 and S18, showing a stronger correlation between low RMSD and better docking score. This is true for both the ponatinib and imatinib applications in DDR1 and Abl1. That also seems true for the trimmed results for Src kinase in Figure S19. I was also surprised that there are structures with very large RMSD but docking scores comparable to the best structures of the lowest RMSD. Might something be done to make the docking score a more effective discriminator?

The docking algorithm and docking score are used to filter out highly improbable docking poses. False positives in predicted docking poses are a common issue across all docking methods as described for instance in:

Fan, Jiyu, Ailing Fu, and Le Zhang. "Progress in molecular docking." Quantitative Biology 7 (2019): 83-89.

Ferreira, R.S., Simeonov, A., Jadhav, A., Eidam, O., Mott, B.T., Keiser, M.J., McKerrow, J.H., Maloney, D.J., Irwin, J.J. and Shoichet, B.K., 2010. "Complementarity between a docking and a high-throughput screen in discovering new cruzain inhibitors." Journal of medicinal chemistry, 53(13), pp.4891-4905.

Moreover, there is always a trade-off between docking accuracy and computational cost. While employing more accurate docking methods may decrease false positives, it can also be resource-intensive. In such scenarios, our approach to enriching holo-structures can be impactful by reducing the number of pocket structures in the input ensembles and significantly enhancing docking efficiency.

(3) I think that it is fine to identify one structure as "IFD winner" but also feel that its significance is overstressed, especially given that it can be identified only in a retrospective analysis rather than through de novo prediction.

We agree with the reviewer. We did not intend to emphasize the specific structure "IFD winner". Rather, we aimed to demonstrate that our method can enrich promising candidates for holo-structures. We verified this by showing that our holo-structure candidates performed well in retrospective docking using IFD, which we previously referred to as "IFD winner". We have now revised this term to "holo-model".

Minor Pointsp. 3 "DymanicBind" should be "DynamicBind"p. 3 Change "We chosen" to "We have chosen" or "we chose."p. 3 In identifying the Schrödinger software Glide and IFD, I recommend removing the subjective modifier "industry-leading."

Modifications done.

**Reviewer #2 (Recommendations For The Authors):**
In the view of this reviewer, the writing is 'choppy'.

We have tried to improve the writing.

**Reviewer #3 (Recommendations For The Authors):**
(1) In Figure 1, the workflow labels (i) to (iv) are not shown on the figures, making it difficult for readers to follow. Consider adding these labels to the figures.

Modifications done.

(2) Explain how Boltzmann ranks were calculated based on unbiased MD simulations to guide the enrichment of holo-like structures in metastable states.

The Methods section is now updated for clarification.

(3) The authors could clarify how the classical DFG-out decoys in the DDR1 rMSA AF2 ensemble are transferred to Abl1 kinase in the Methods section.

The Methods section is now updated for clarification.

(4) The authors can clarify the methodology section by providing more detailed explanations about how the unbiased MD simulations are performed, including which MD simulation software was used and whether energy minimization and equilibrium steps were needed as in conventional MD simulations, and other setup details.

The Methods section is now updated for clarification.

(5) The validation of the proposed approach in this work used three kinase proteins. The authors can enhance the discussion section by addressing other types of protein structure prediction that can use the proposed approach in drug discovery, beyond the three kinase proteins tested.

The proposed approach is theoretically applicable to other types of proteins, such as GPCRs, where both conformational selection and the induced-fit effect are crucial. We have expanded the discussion on the generalization of our protocol in the Conclusion section.

(6) The authors should add appropriate citations for the software and tools used in the manuscript. For example, a reference should be added for the Glide XP docking experiments that utilized the Maestro software. Double-check all related software citations.

We have now updated the citations for docking experiments based on the instruction of the Maestro Glide User manual and IFD User manual.

(7) The authors should consider offering a comprehensive list of software tools and databases utilized in the study to assist in replicating the experiments and further validating the results.

We have now added a summary of tools used in the Methods section.